# Neurophysiological Correlates of a Single Session of Prefrontal tDCS in Patients with Prolonged Disorders of Consciousness: A Pilot Double-Blind Randomized Controlled Study

**DOI:** 10.3390/brainsci10070469

**Published:** 2020-07-21

**Authors:** Manon Carrière, Sepehr Mortaheb, Federico Raimondo, Jitka Annen, Alice Barra, Maria C. Binda Fossati, Camille Chatelle, Bertrand Hermann, Géraldine Martens, Carol Di Perri, Steven Laureys, Aurore Thibaut

**Affiliations:** 1Brain2 Clinic, Coma Science Group, GIGA Consciousness, University and University Hospital of Liège, 4000 Liège, Belgium; s.mortaheb@uliege.be (S.M.); fraimondo@uliege.be (F.R.); jitka.annen@uliege.be (J.A.); a.barra@uliege.be (A.B.); mariachiara.bindafossati@gmail.com (M.C.B.F.); camillechatelle@gmail.com (C.C.); geraldine.martens@uliege.be (G.M.); diperric@gmail.com (C.D.P.); steven.laureys@uliege.be (S.L.); athibaut@uliege.be (A.T.); 2Institut du Cerveau et de la Moelle épinière, ICM, PICNIC Lab, F-75013 Paris, France; bertrand.hermann@aphp.fr; 3ICU, Hôpital Européen Georges Pompidou, APHP, Université de Paris, F-75013 Paris, France

**Keywords:** minimally conscious state, neuromodulation, non-invasive brain stimulation, electrophysiology, brain injury

## Abstract

**Background.** Transcranial direct current stimulation (tDCS) over the left dorsolateral prefrontal cortex (lDLPFC) was reported to promote the recovery of signs of consciousness in some patients in a minimally conscious state (MCS), but its electrophysiological effects on brain activity remain poorly understood. **Objective.** We aimed to assess behavioral (using the Coma Recovery Scale—Revised; CRS-R) and neurophysiological effects (using high density electroencephalography; hdEEG) of lDLPFC-tDCS in patients with prolonged disorders of consciousness (DOC). **Methods.** In a double-blind, sham-controlled, crossover design, one active and one sham tDCS (2 mA, 20 min) were delivered in a randomized order. Directly before and after tDCS, 10 min of hdEEG were recorded and the CRS-R was administered. **Results.** Thirteen patients with severe brain injury were enrolled in the study. We found higher relative power at the group level after the active tDCS session in the alpha band in central regions and in the theta band over the frontal and posterior regions (uncorrected results). Higher weighted symbolic mutual information (wSMI) connectivity was found between left and right parietal regions, and higher fronto-parietal weighted phase lag index (wPLI) connectivity was found, both in the alpha band (uncorrected results). At the group level, no significant treatment effect was observed. Three patients showed behavioral improvement after the active session and one patient improved after the sham. **Conclusion.** We provide preliminary indications that neurophysiological changes can be observed after a single session of tDCS in patients with prolonged DOC, although they are not necessarily paralleled with significant behavioral improvements.

## 1. Introduction

Therapeutic options for severely brain-injured patients with disorders of consciousness (DOC) are limited and need to be improved in order to influence long-term outcomes. Among the non-invasive brain stimulation techniques, transcranial direct current stimulation (tDCS) has shown promising results in DOC [1]. This technique modulates the excitability of targeted brain regions by inducing a weak electrical current (usually 1–2 mA) between two electrodes (an anode and a cathode) placed on the scalp [2]. Different montages have been tested in this population, but the most effective target so far is the prefrontal cortex. tDCS applied over the left dorsolateral prefrontal cortex (lDLPFC) has been shown to improve behavioral responsiveness in patients in a minimally conscious state (MCS), either after a single stimulation [3] or repeated sessions [4,5], as measured by the Coma Recovery Scale—Revised (CRS-R) [6]. About half of the patients in MCS (i.e., showing fluctuating but reproducible signs of consciousness) seem to be responsive to this technique, while no treatment effects have been observed in patients with unresponsive wakefulness syndrome (UWS, i.e., showing only reflexive behaviors) [7,8]. According to two recent studies, the presence of minimal residual brain functioning might be a prerequisite to benefit from tDCS [9,10]. Higher metabolic activity and gray matter volume as well as higher theta network centrality are indeed observed in lDLPFC-tDCS responders (i.e., showing one new sign of consciousness after active tDCS only) compared to non-responders [9,10].

To date, only a few studies have investigated tDCS-induced EEG changes in patients with prolonged DOC, and only one used a randomized controlled double-blind design. Cortical excitability changes following lDLPFC tDCS were assessed by Bai et al. in nine UWS patients and seven MCS patients using transcranial magnetic stimulation coupled with EEG [11]. Compared to baseline and sham, active tDCS induced global cerebral excitability increases in early time windows (0–100 and 100–200 ms) in MCS patients, while in UWS patients, global cerebral excitability increases were observed in the 0–100 ms interval but decreased in the 300–400 ms interval. The local cerebral excitability was significantly different between MCS and UWS patients. The same authors later found, in a sham-controlled crossover study using resting-state EEG, increased fronto-parietal coherence in the theta band and a decrease in the gamma band in MCS (n = 8) patients following lDLPFC tDCS, while no changes were observed in UWS (n = 9) [12]. Another study found decreased central-parietal coherence in the delta band following high-density tDCS over the posterior parietal cortex, along with increased CRS-R total scores in 9 out of 11 DOC patients, even though no significant treatment effect was reported [13]. A sham-controlled, randomized, double-blind study showed a significant clinical improvement in MCS patients (n = 8) following ten sessions of active tDCS targeting the lDLPFC, along with an increase in P300 amplitude [14]. After two weeks of active tDCS over the lDLPFC, Cavinato et al. found an increase in the power and coherence of the frontal and parietal alpha and beta frequency bands, along with clinical improvements in MCS patients (n = 12), whereas UWS (n = 12) only showed some local changes in the slow frequencies over frontal electrodes [15]. Finally, Hermann et al. recently reported increased power and connectivity in the theta-alpha band after one session of lDLPFC tDCS in responders (12/60–20%; 7/32 MCS–21.9%) in a prospective open-label study with DOC patients [16]. In this pilot study, we aim to verify these findings on theta and alpha bands by investigating the direct effects of a single session of tDCS applied over the lDLPFC at the behavioral and neurophysiological levels in DOC patients, using a double-blind, randomized, controlled, crossover design.

## 2. Materials and Methods

### 2.1. Patients

We prospectively enrolled medically stable patients with DOC, who had been hospitalized for one week for a multimodal assessment of their level of consciousness between January 2018 and March 2019. Inclusion criteria were (1) diagnosed as MCS or emergent from the MCS (EMCS) following an acquired traumatic or non-traumatic injury by means of repeated (≥5) CRS-R and (2) more than 3 months post-injury. We excluded patients in UWS and patients with any contraindications to tDCS (i.e., uncontrolled seizure, implanted electronic brain medical device or pacemaker, history of recent craniotomy or cranioplasty in the frontal region) or with premorbid neurological or psychiatric conditions (see Figure 1 for a flowchart). Written informed consent was obtained by the legal representative. The study was registered (ClinicalTrials.gov NCT03823508) and approved by the local ethics committee, and is reported in accordance with CONSORT guidelines.

### 2.2. Data Acquisition

In a double-blind, sham-controlled, crossover design, anodal and sham tDCS were allocated in a randomized order and administered in two sessions separated by at least 48 h. Two experienced investigators were involved in data collection. One investigator was in charge of the tDCS application, whereas the other performed the CRS-R assessments. For each patient, the first experimenter received two blinded codes from a third person, one for the active and one for the sham stimulation, allowing the investigator (and the patients) to be completely blind regarding the treatment allocation.

Our primary outcome measures were the neurophysiological alterations, as measured by ten minutes of resting-state hdEEG recorded with eyes open directly before and after the stimulation/sham. As a secondary outcome, behavioral improvement was assessed by means of CRS-R assessments performed by an experienced experimenter directly before and after the tDCS/sham (see Figure 2 for the protocol).

*tDCS:* We used the NeuroConn DC Stimulator Plus (NeuroConn Company Ltd.: Ilmenau, Germany). This device offers a built-in placebo mode for the sham session, which is activated by an anonymous code number and includes ramp periods at the beginning and the end of sham stimulation to mimic the feeling of active tDCS. Direct current was applied by a battery-driven constant current stimulator using saline-soaked surface sponge electrodes (7 × 5 cm) with the anode placed over the lDLPFC (F3 according to the 10–20 international system for EEG placement) [17] and the cathode positioned on the right supraorbital region, as previously described [3]. The stimulation had an intensity of 2 mA and lasted 20 min, with 30 s of ramp up at the beginning and 30 s of ramp down at the end of the stimulation.

*EEG:* hdEEG recordings were acquired with a 256-channel saline-solution compatible electrode net (Electrical Geodesics, EGI) with a sampling rate of either 250 Hz or 500 Hz.

*CRS-R:* The CRS-R consists of 23 hierarchical items (from reflexes, e.g., blinking to threat or auditory startle; to more complex voluntary behaviors, e.g., command following, visual pursuit) organized into six subscales assessing auditory, visual, motor, oromotor, communication, and arousal functions [6]. The items are hierarchically organized within each subscale, the lowest items representing reflexes whereas the highest represent consciousness-related behaviors.

### 2.3. Data Analysis

#### 2.3.1. EEG

##### Preprocessing

Signals with sampling rates of 500 Hz were down-sampled to 250 Hz. All data were filtered first using an IIR 8th order Butterworth lowpass filter with a cut-off frequency of 45 Hz and then an IIR 4th order Butterworth high pass filter with cut-off frequency of 0.5 Hz. To minimize the effect of electric line noise, a notch filter was applied on the 50 Hz frequency. Bad electrodes (i.e., flat-line or highly artifactual electrodes) were identified by visual inspection and excluded. Data were segmented into 2 s epochs with a random jitter (50 to 550 ms) between epochs, and then bad epochs (i.e., epochs with high frequency noise and/or very high amplitude variation) were identified by visual inspection and rejected. Independent components were computed using the infomax algorithm [18], and the components explaining 99% of the data variance were selected for further analysis. Independent Component Analysis (ICA) components were visually inspected using power spectral density, spatial distribution, and variance distribution over epochs. Components representing non-neural activity (mainly related to the eye blink, heartbeat, and muscle artifacts) were set to zero, and the remaining components were used to reconstruct the EEG signals. The removed bad channels were interpolated using spherical interpolation and all electrodes were re-referenced to their average. Electrodes of the neck and cheeks were neglected for subsequent analysis, and 224 electrodes of the scalp were kept. Mean and standard deviation of number of interpolated channels and retained epochs for each recording session are as follows, respectively: pre-sham (30.1 ± 18.2, 193.9 ± 51.4), post-sham (26.2 ± 13.8, 151.6 ± 58.1), pre-active (21.9 ± 13.5, 183.8 ± 48.0), and post-active (27.3 ± 21.4, 175.5 ± 50.0).

##### Spectral Power Analysis

For each session, the power spectral density (PSD) was first estimated for each epoch and each channel using Welch’s method, with 4096 samples per segment and a 200-sample overlap using a 256-sample Hanning window. Then, the power of delta (1–4 Hz), theta (4–8 Hz), alpha (8–12 Hz), and beta (12–30 Hz) was computed using summation of power values over frequency bins. The relative power was calculated at each frequency band by dividing its power by the total power of the broadband (i.e., 0.5–45 Hz) signal. Then, the trimmed mean (20% trim) of the relative power of each frequency band was calculated over the epochs for each electrode. To study the treatment effect of tDCS on spectral power, changes in power values after stimulation in sham and active sessions were analyzed ((post active–pre active) vs. (post sham–pre sham)) at the group level using a related *t*-test, and to overcome the multiple comparison problem, a non-parametric cluster permutation test was performed [19]. For the subject level analysis, the ANOVA model was fitted to the single-epoch power values of each electrode at different frequency bands, considering session (sham vs. active) and period (pre vs. post) as the factors of the model. In this regard, for each subject and each frequency band, the effects of session and period and their interaction were tested. Again, to solve the multiple comparison issue, a cluster permutation test was performed. For all cluster permutation tests, the number of permutations was set to 1000.

##### Connectivity Analysis

Connectivity analyses were performed using the weighted symbolic mutual information (wSMI) [20] and weighted phase lag index (wPLI) [21]. Before calculation of wSMI, current source density (CSD) estimates were applied to each subject separately to reduce the volume conduction effect and increase the spatial focalization of EEG information [22]. Mutual information was computed in alpha and theta frequency bands since these two frequency bands have been shown to be related to conscious processes in DOC [16,23,24]. The connectivity matrices across electrodes were averaged across epochs. The median connectivity of each electrode to all the other electrodes was computed in order to create connectivity maps. Connectivity between regions of right frontal, left frontal, right central, left central, right parietal, and left parietal was computed by calculating the mean value of connectivity between the electrodes in each pair of regions. Connectivity changes after two sessions of tDCS (sham and active) were analyzed by contrasting changes in active vs. sham conditions ((post active–pre active) vs. (post sham–pre sham)) at the group level using paired *t*-tests.

All preprocessing and processing scripts were written as custom scripts in Python, MNE-Python [25], and NICE [26]. These scripts are publicly available at https://github.com/fraimondo/shock.

#### 2.3.2. Behavioral

Statistical analyses of behavioral data were performed using SPSS version 17.0 [27]. We investigated the presence of a carry-over effect by comparing the CRS-R total scores before active tDCS and before sham tDCS using a Wilcoxon match-paired signed-rank test. In the absence of such an effect, we looked for any treatment effect using the same tests, by comparing the differences in CRS-R total scores as follows: (post active–pre active) vs. (post sham–pre sham), at the group-level and in traumatic brain injuries (TBI) and non-TBI subgroups.

As exploratory analyses, we looked at the difference in CRS-R total scores between pre- and post-active, as well as pre- and post-sham.

## 3. Results

Thirteen patients received both active and sham tDCS in a crossover study design. Two patients were excluded due to missing behavioral data. Finally, eleven patients (three women, mean age 46 ± 14 years, median time since injury = 5 months (min-max = 3–25), three traumatic etiologies, three cardiac arrests, four aneurysms, one meningitis) were included in the study. Based on repeated CRS-Rs, six patients were diagnosed as MCS−, four patients as MCS+, and one patient as EMCS. All MCS patients (n = 10) were included in the group-level behavioral analyses. For the EEG analysis, two subjects were excluded due to highly artifacted or missing data, and EEG group-level analyses were finally performed on nine patients (see Table 1 and Figure 1 for the study flowchart).

### 3.1. EEG Outcome

#### 3.1.1. Spectral Power Analysis

Considering the uncorrected electrode-wise related *t*-test results, we can observe an increase in relative power in the frontal, parietal and occipital regions for the theta band, and an increase in relative power mainly in the central regions for the alpha band (see Figure 3). After correction for multiple comparisons using permutation and cluster-level correction, no significant results were observed. At the subject level, we observed spectral power variability in both sham and active sessions for all subjects in all frequency bands. The cluster permutation test on the F statistics of the ANOVA model found a significant effect of the period, the session, and their interaction on the power values of all subjects at each frequency band (Figure A1, Figure A2, Figure A3 and Figure A4. show topography maps and their corresponding analysis results for all subjects in more detail). The results show high variability in EEG recordings in both short time periods (pre vs. post (i.e., 20 min gap)) and long time periods (sham vs. active (i.e., at least 48 h gap)) at the subject level.

#### 3.1.2. Connectivity Analysis

For connectivity analysis, two connectivity metrics (wPLI and wSMI) were calculated in theta and alpha frequency bands. After correcting for multiple comparisons, no significant change was observed in any of the frequency bands. However, in uncorrected statistics, an increase in wSMI alpha connectivity was observed in the parietal region (i.e., between right parietal and left parietal ROIs), and an increase in wPLI alpha connectivity was observed in the fronto-parietal regions (i.e., between right frontal and left parietal ROIs; see Figure 4). All the *p*-values obtained for these analyses are available in Table A1, Table A2, Table A3 and Table A4.

### 3.2. Behavioral Outcome.

At the group-level (n = 10), the median (IQR) CRS-R total score before active stimulation was 8 (6–10.5) and 8 (6–9.5) after active stimulation. For the sham condition, this was 8.5 (7–12.25) before and 7.5 (4.75–12.5) after. No carry-over effect was identified (Z = −1.1, *p* = 0.27) and we did not find any significant treatment effect either (Z = −0.36, *p* = 0.72). No significant differences in CRS-R total scores were found between pre- and post-active (Z = −1.39; *p* = 0.166) or between pre- and post-sham (Z = −1.27; *p* = 0.203). When subcategorizing by etiology, we did not find any significant treatment effect for TBI (Z = −1.63, *p* = 0.10) or non-TBI patients (Z = −1.06, *p* = 0.28).

At the individual level, three patients presented improvement following the active but not the sham session. The first patient (S06) is a 40-year-old male who suffered a TBI two years prior. He was diagnosed as MCS+ and he showed object localization only after the active tDCS. The second patient (S04) is a 59 year-old male who had a cardiac arrest 11 months before tDCS. He was diagnosed as MCS- and showed for the first time localization to sound only after the active tDCS. The third patient (S07) is a 63-year-old male diagnosed as MCS+ five months after a TBI, who showed intentional communication only after the active tDCS. None of these new items resulted in a change in diagnosis.

We also found six patients who had a lower total score at the post-active CRS-R than at the pre-active CRS-R; nonetheless, none of them lost a sign of consciousness. Five patients obtained a lower total score at the post-sham CRS-R compared to the pre-sham CRS-R. Table 2 shows the CRS-R totals and sub-scores obtained by each patient and for each session.

## 4. Discussion

The aim of this pilot study was to explore the direct effects of a single session of tDCS targeting the lDLPFC on the behavioral and neurophysiological outcomes of patients with severe brain injury in a randomized, controlled, double-blind trial. Our preliminary findings show that tDCS tends to modulate the brain activity and connectivity of the stimulated area but also reaches more distant regions in patients with prolonged DOC, even in the absence of behavioral improvement.

Considering spectral power analysis, we observed an increase in theta power in the frontal and posterior regions and an increase in alpha power mainly in central regions. However, these results did not survive multiple comparisons correction for the number of electrodes. This is certainly due to the small number of subjects, which leads to insufficient statistical power to reject the null hypothesis but cannot prove that the null hypothesis can be accepted. However, the present findings are in line with previous neurophysiological studies investigating the correlation between EEG metrics and level of consciousness. Normalized theta and alpha power has been shown to efficiently index the states of consciousness [23]. In addition, increases in theta and alpha bands in the central and posterior regions were also reported in DLPFC-tDCS responders compared to non-responders [16]. Our results are therefore in line with the existing literature, showing that increases in alpha and theta power may reflect a higher level of brain activity following active tDCS, even if it is not translated into behavioral responsiveness in all patients after a single session. Single subject analysis showed that there was a high variability in spectral power after both active and sham sessions and even for the interaction of period (i.e., pre-stimulation vs. post-stimulation) and session (i.e., sham vs. active) for all subjects in all frequency bands. The significant interaction of period and session shows that this variability can be linked to the application of tDCS in the patients. In addition, this high individual variability could be the reason that we did not see significant results in the group level analysis.

To explore connectivity, we used wSMI and wPLI, which have been shown to be robust metrics against volume conduction ([18,19], respectively). These two metrics, however, account for different types of functional interactions [28]. Indeed, wPLI would be sensitive to a mixture of linear and non-linear interdependencies (mainly simple linear connectivity), while purely non-linear and complex inter-areal coupling dynamics of the brain can be captured by wSMI. Although uncorrected, our results showed an increase in wSMI connectivity in the alpha band between the right and left parietal regions and an increase in fronto-parietal wPLI connectivity in the alpha band as well. Previous studies have shown that these two metrics in the alpha band correlate with DOC patients’ levels of consciousness [23,24]. Chennu et al. highlighted that the re-emergence of behavioral awareness is correlated with the fronto-parietal alpha band wPLI connectivity. In addition, other studies showed that a decrease in the posterior-anterior alpha band’s wPLI connectivity was associated with the transition into unconsciousness due to sedation or sleep [29,30,31]. Based on these previous results, and considering our findings showing that wPLI connectivity in the alpha band was decreased in the sham and increased in the active sessions (which led to a significant increase in fronto-parietal alpha band connectivity), we may argue that anodal tDCS deprived patients of drowsiness in the active session.

The fact that we found an increase in parieto-parietal wSMI and fronto-parietal wPLI connectivity, and considering their ability to capture purely non-linear and a mix of linear and nonlinear interactions, respectively, might be explained by a recent theory according to which posterior parts (including parietal regions) of the brain are critical for consciousness, even if fronto-parietal connectivity is still determinant [32]. These authors indeed questioned the role of the frontal cortex in supporting consciousness, suggesting a direct contribution to specific contents of experience from posterior parietal regions, whereas frontal regions rather modulate the neural correlates of consciousness but would not directly contribute to consciousness. In the present study, we found that, even if tDCS was applied over the prefrontal cortex, both fronto-parietal connectivity and connectivity between the right and left parietal regions were modulated. This is in line with a functional MRI study on healthy subjects, according to which tDCS would not only modulate the activity under the stimulated area but also brain network connectivity encompassing long distance brain regions [33]. In DOC patients, the hypothesis is that some long-distance connectivity (encompassing cortical or subcortical regions) may be chronically under-active and that the action of tDCS on cerebral activity and connectivity might counteract this under-activation and induce clinical improvements [10]. tDCS-EEG randomized controlled trials including a convenient number of patients (e.g., 58—see previous paragraph) or longer periods of stimulation (e.g., 20 sessions of tDCS)—expecting a significant behavioral effect at the group level—need to be conducted in order to determine whether the fronto-parietal and the inter-hemispheric parietal connectivity underlie the behavioral improvement of consciousness, in favor of the posterior “hot-spot” [32].

Previous studies have also reported that theta band connectivity is affected by tDCS [16,20]. Although not significant (probably due to small sample size), theta band wSMI connectivity showed an increase in centro-parietal and fronto-parietal in this band, which is in line with the results of Hermann et al. [16], who found an increase in centro-parietal wSMI connectivity in the theta band, mainly in tDCS responders, in an open label study. Our present findings tend to confirm Hermann’s results, using a sham controlled crossover design, highlighting the role of posterior associative cortices and long-range connectivity, especially in the theta band, in consciousness recovery [20].

Behaviorally, we did not find any significant treatment effect. Nevertheless, three patients displayed a new behavior associated with consciousness according to the CRS-R only after the active tDCS, which falls within the range (21.9–43%) found by previous studies after one single session of tDCS [3,16]. The absence of a significant treatment effect at the group level might be attributed to several reasons. Firstly, and as previously shown in DOC and in other pathologies, sometimes one session is not sufficient to induce clinical enhancement and more sessions are necessary [5,34,35,36]. This is especially true in DOC patients who present particularly serious and extensive brain lesions. Secondly, we applied the tDCS over the lDLPFC because it seems to be the most effective area to target, according to previous studies. However, we did not examine the localization and extent of the patients’ lesions before including them, and perhaps the lDLPFC was not the most adequate area to target for all our patients (see Table 1 for a summary of patients’ main structural lesions). As a matter of fact, we know that a minimum preservation of the stimulated areas is required in order to respond to tDCS [9]. Finally, our study may be under-powered. Given the treatment effect at the group level, we calculated an effect size of 0.29, meaning that, to reach a power of 80%, we would have needed 58 patients.

Some limitations in this study should be acknowledged. The first limitation is undoubtedly the small size of our sample. We only included patients who were medically stable and did not present any contraindication to tDCS (e.g., uncontrolled seizure, implanted electronic brain medical device or pacemaker, recent craniotomy or cranioplasty in the frontal region), corresponding to 54% of the total number of patients that came to our hospital during the recruitment period (see Figure 1 for the flow chart). In this pilot study, we aimed to explore the neurophysiological effect of a single session of prefrontal tDCS; however, the present findings cannot be generalized. Multi-center trials are therefore needed in order to overcome the challenge that represents the recruitment of DOC patients.

Secondly, our protocol consisted of only a single session of tDCS over the lDLPFC and we know that repeated tDCS sessions are more effective to induce significant behavioral changes [4,34,37]. We can reasonably assume that repeated tDCS sessions would induce stronger neurophysiological changes than a single session. However, repeated assessments were not feasible, as recruited patients were hospitalized for a few days only and were also assessed with other clinical and neuroimaging examinations. Finally, there might be differences according to etiologies that we could not investigate due to our limited sample. Future studies with larger samples should aim to explore the effect of etiology on neurophysiological correlates of tDCS in DOC.

## 5. Conclusions

The results of this pilot study tend to show that a single session of active tDCS over the lDLPFC modulates brain activity and connectivity in patients with severe brain injury, particularly in the theta and alpha frequency bands, in line with previous EEG studies evaluating the correlation between brain connectivity and the level of consciousness. Based on our preliminary results, tDCS tends to influence fronto-parietal and inter-hemispheric parietal connectivity. Larger randomized controlled trials need to be conducted to confirm that these parameters are correlated with the improvement of consciousness following tDCS application. We also provide preliminary indications that hdEEG seems to be sensitive to quantifying tDCS-induced changes in brain activity and connectivity (at the local and global level), more so in a pathological population of severely brain injured patients. These neurophysiological changes occurred in the absence of behavioral improvement at the group level, as quantified with the CRS-R total scores. The results of this pilot, randomized, double-blind, sham-controlled study are encouraging and highlight the potential of tDCS to induce neurophysiological effects in a population of severely brain injured patients.

## Figures and Tables

**Figure 1 brainsci-10-00469-f001:**
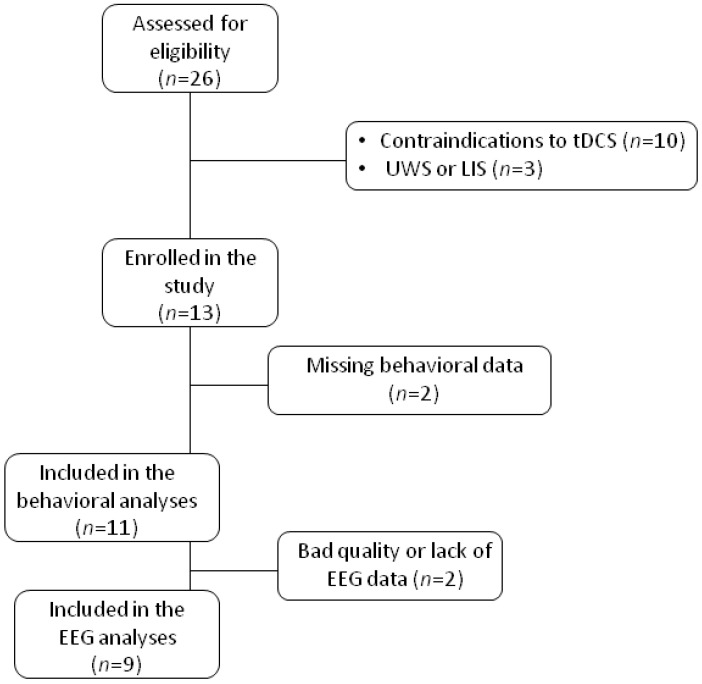
Study flowchart. tDCS = transcranial direct current stimulation; UWS = unresponsive wakefulness syndrome; LIS = locked-in syndrome; EEG = electroencephalography.

**Figure 2 brainsci-10-00469-f002:**
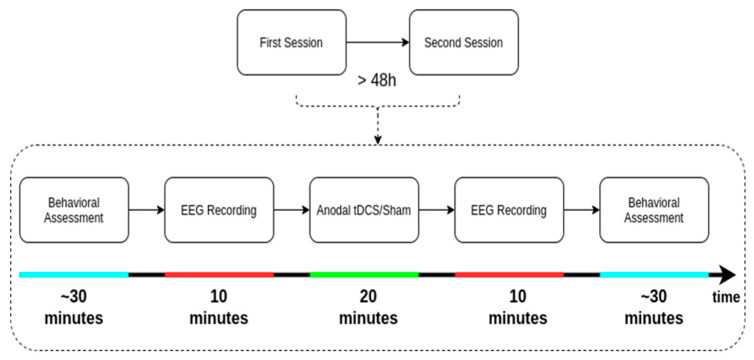
Protocol of the study. CRS-R = Coma Recovery Scale—Revised; EEG = electroencephalography; tDCS = transcranial direct current stimulation.

**Figure 3 brainsci-10-00469-f003:**
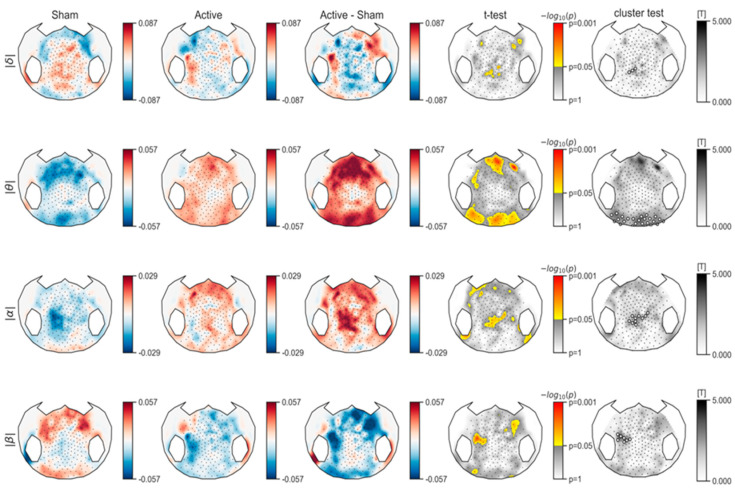
Relative power differences in delta, theta, alpha, and beta bands in sham and active sessions as well as for active session compared to sham session. Uncorrected related *t*-test results are shown for a significance level of 0.05. In the permutation cluster test results, white circles show the electrodes of the clusters found in the test. Significant clusters are shown in red (no significant cluster in this figure).

**Figure 4 brainsci-10-00469-f004:**
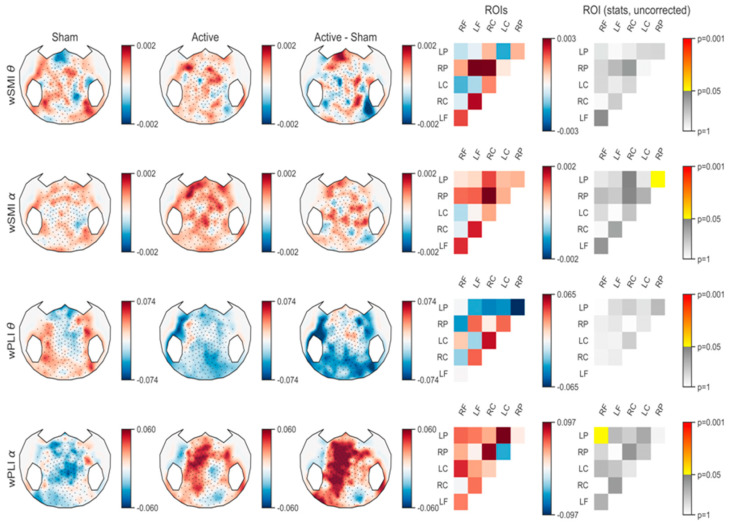
Connectivity differences in sham and active tDCS and active compared to sham sessions.

**Table 1 brainsci-10-00469-t001:** Demographic data of the eleven patients included in the study.

Patient	Gender	Age	Etiology	Structural Brain Lesions (MRI)	Time Since Injury (months)	Behavioral Analysis	EEG Analysis
S01	M	34	TBI	Mature contusion damage extensively involving the left hemisphere, severe in the left frontal, (lateral and medial) temporal, and mesial parietal-occipital lobes	5	X	X
S02	M	45	CA	Cortico-subcortical atrophy. Signs of rolandic and parieto-occipital ischemia (bilaterally)	10		X
S03	M	52	CA	Diffuse signal abnormality involving the periventricular white matter and and fronto-parieto-temporal and occipital cortex and subcortical white matter, mainly in the right hemisphere	10	X	X
S04	M	59	CA	Diffuse and extensive white matter abnormality involving both hemispheres and severe and extensive cortico-subcortical atrophy	11	X	X
S05	M	31	Meningitis	Diffuse signal abnormality affecting the temporal, frontoinsular, and frontobasal lobes bilaterally	3	X	X
S06	M	40	TBI	Sequelae of hemorrhagic contusion in the left midbrain and diffuse signal abnormality affecting the left frontal lobe	25	X	X
S07	M	63	Aneurysm	Mature midbrain damage, diffuse periventricular and frontal white matter abnormality, and diffuse cortico-subcortical atrophy	5	X	X
S08	M	23	TBI	Established contusion lesions in the right frontal and right mesial temporal lobes and left basal ganglia	3	X	X
S09	F	47	Aneurysm	Left temporo-occipital encephalomalacia with partial confluence with the left lateral ventricle associated with diffuse surrounding signal abnormality	4	X	X
S10	F	71	Aneurysm	Sequelae of hemorrhagic infarcts in the right mesial occipital and left mesial frontal lobes	13	X	
S11	F	50	Aneurysm	Diffuse and extensive white matter signal abnormalities in the fronto-parietal-insular and temporal lobes, mainly affecting the left hemisphere	3	X	

UWS = unresponsive wakefulness syndrome; MCS = minimally conscious state; M = male; F = female; CA = cardiac arrest; TBI = traumatic brain injury.

**Table 2 brainsci-10-00469-t002:** Behavioral outcomes. Coma Recovery Scale—Revised totals and sub-scores for the pre- and post-sham and the pre- and post-active sessions of transcranial direct current stimulation. The CRS-R total scores in bold and underlined correspond to the total scores obtained by the three responders before and after active tDCS.

Patient	tDCS Allocation	Pre-Sham CRS-R	Post-Sham CRS-R	Pre-Active CRS-R	Post-Active CRS-R
S01	Active/sham	8 (1-3-2-0-0-2)	8 (1-3-2-0-0-2)	9 (1-3-1-2-0-2)	8 (1-3-1-1-0-2)
S03	Sham/active	9 (1-3-1-2-0-2)	5 (1-1-1-1-0-1)	8 (1-3-1-1-0-2)	7 (1-3-1-1-0-1)
S04	Sham/active	7 (1-2-2-1-0-1)	7 (1-3-0-1-0-2)	6 (1-3-0-1-0-1)	7 (2-1-0-2-0-2)
S05	Sham/active	11 (0-3-5-2-0-1)	10 (0-3-5-1-0-1)	12 (0-3-5-2-0-2)	8 (0-1-5-1-0-1)
S06	Active/sham	9 (0-3-2-2-0-2)	12 (3-3-2-2-0-2)	9 (1-3-2-1-0-2)	10 (1-4-2-1-0-2)
S07	Active/sham	16 (4-5-5-2-0-0)	14 (3-4-5-2-0-0)	8 (3-3-0-2-0-0)	9 (3-3-0-2-1-0)
S08	Sham/active	18 (4-4-5-2-1-2)	17 (4-5-5-2-0-2)	19 (4-5-4-2-2-2)	17 (4-5-4-0-2-2)
S09	Sham/active	7 (1-1-2-2-0-1)	7 (1-1-2-2-0-1)	6 (0-1-2-1-0-2)	5 (1-1-1-1-0-1)
S10	Sham/active	8 (3-1-1-1-0-2)	3 (0-0-2-1-0-0)	3 (0-0-2-1-0-0)	2 (0-0-1-1-0-0)
S11	Sham/active	2 (0-1-0-0-0-1)	4 (0-3-0-0-0-1)		

Abbreviations: MCS = minimally conscious state.

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
