# Peer review of "Neurophysiological Correlates of a Single Session of Prefrontal tDCS in Patients with Prolonged Disorders of Consciousness: A Pilot Double-Blind Randomized Controlled Study"

_brainsci, 2020, doi:10.3390/brainsci10070469_

Round 1

Reviewer 1 Report

Neurophysiological correlates of a single session of prefrontal tDCS in patients with Prolonged Disorders of consciousness: a Pilot Double-Blind Randomized Controlled Study

This pilot study evaluated the effects of transcranial direct current stimulation (tDCS) over the left dorsolateral prefrontal cortex (lDLPFC) brain activity as measured by quantitative EEG along with behavioral assessment. The study found increased alpha-band activity in the central region and theta-band activity in the frontal and parietal areas after single-session tDCS.

Considering the difficulty of recruiting homogenous patients with a minimally conscious state (MCS) and the even greater difficulty in applying tDCS and quantitative EEG in those patients, this study seems brave and inspiring.

Although single-session tDCS did not show any clinical effects, the changes in brain activity identified after tDCS appears to be a valuable result in that it suggests the possibility that a more advanced method of brain stimulation will be effective in restoring consciousness in patients with MCS.

There are several issues that need to be edited before considering publication.

  1. Although the primary outcome in this study was a change in brain activity after tDCS and the behavioral effects from tDCS were a secondary parameter, the results section focuses more on the latter than the former. The primary outcome needs to be placed first.
  2. Brief descriptions of the meanings of alpha and theta bands related to consciousness could help readers better understand this article.
  3. In the Discussion section, it is recommended to add a description of how tDCS affects brain activity and connectivity, leading to changes in alpha and theta power and connectivity in these patients. The authors also need to add their theory on how single-session tDCS over the left prefrontal cortex changed brain activity and connectivity even in remote areas.
  4. As the authors already noted, there was no significant difference in brain activity when correcting for multiple comparisons at the group level. This is the main weakness of this study. The authors used a cluster permutation test model at the subject level and found significant effects in the period, session, and interaction. To justify the statistical method, please discuss whether the statistician performed data analysis as an acknowledgement or in the Method section.
  5. Please check the sentence on lines 291–293 of page 10. Specifically, check to see whether this is an error on line 292: “…following tDCS, after the active tDCS…”

Author Response

Reviewer #1

This pilot study evaluated the effects of transcranial direct current stimulation (tDCS) over the left dorsolateral prefrontal cortex (lDLPFC) brain activity as measured by quantitative EEG along with behavioral assessment. The study found increased alpha-band activity in the central region and theta-band activity in the frontal and parietal areas after single-session tDCS. Considering the difficulty of recruiting homogenous patients with a minimally conscious state (MCS) and the even greater difficulty in applying tDCS and quantitative EEG in those patients, this study seems brave and inspiring. Although single-session tDCS did not show any clinical effects, the changes in brain activity identified after tDCS appears to be a valuable result in that it suggests the possibility that a more advanced method of brain stimulation will be effective in restoring consciousness in patients with MCS.

We thank the Reviewer for these positive and encouraging comments.

There are several issues that need to be edited before considering publication.

  1. Although the primary outcome in this study was a change in brain activity after tDCS and the behavioral effects from tDCS were a secondary parameter, the results section focuses more on the latter than the former. The primary outcome needs to be placed first.

We thank the Reviewer for this suggestion. The primary outcome (EEG) has now been placed first within the Methods and Results sections.

  1. Brief descriptions of the meanings of alpha and theta bands related to consciousness could help readers better understand this article.

We agree with the reviewer that this is an important point to discuss. Alpha and theta bands have been studied extensively in researches about consciousness. Different signal features in these frequency bands were suggested as the signatures of the states of consciousness. For example, fronto-parietal connectivity in the alpha band has been suggested as a potential correlate of the level of consciousness (Chennu et al, 2017). In addition, it has been suggested that the fronto-parietal networks mediate a serial stream of conscious states at theta-like frequencies (Sitt et al, 2014). Even considering power spectrum density of the EEG signal, power in these two frequency bands in the parietal region has been shown to be the most discriminative feature between MCS and UWS patients (Sitt et al, 2014).

The results of these abovementioned studies highlighting the relation between alpha and theta bands and consciousness were already presented in the Discussion section of the manuscript.

Page 9 reads: « Normalized theta and alpha power has been shown to efficiently index the states of consciousness [23]. In addition, increases in theta and alpha bands in central and posterior regions were also reported in DLPFC-tDCS responders compared to non-responders [16]. Our results are therefore in line with the existing literature, showing that increases of alpha and theta power may reflect a higher level of brain activity following active tDCS, even if it is not translated into behavioral responsiveness in all patients after a single session. »

« Previous studies have shown that these two metrics in the alpha band correlate with DOC patients’ level of consciousness [23,24]. Chennu et al highlighted that re-emergence of behavioral awareness is correlated with the fronto-parietal alpha band wPLI connectivity. In addition, other studies showed that a decrease of posterior-anterior alpha band wPLI connectivity was associated with transition into unconsciousness due to sedation or sleep [29–31]. »

  1. In the Discussion section, it is recommended to add a description of how tDCS affects brain activity and connectivity, leading to changes in alpha and theta power and connectivity in these patients. The authors also need to add their theory on how single-session tDCS over the left prefrontal cortex changed brain activity and connectivity even in remote areas.

We agree with the reviewer that these points are important and need to be discussed in the manuscript. However, there were already described in the introduction and the discussion (see below).

-Regarding how tDCS affects brain activity and connectivity in these patients, page 2 (introduction) reads: « To date, only a few studies have investigated tDCS-induced EEG changes in patients with prolonged DOC, and only one used a randomized controlled double-blind design. Cortical excitability changes following lDLPFC tDCS were assessed by Bai et al in nine UWS patients and seven MCS patients using Transcranial Magnetic Stimulation coupled with EEG [11]. Compared to baseline and sham, active tDCS induced global cerebral excitability increases in early time windows (0–100 and 100-200ms) in MCS patients, while in UWS patients global cerebral excitability increases were observed in the 0-100ms interval but decreased in the 300-400 ms interval. The local cerebral excitability was significantly different between MCS and UWS patients. The same authors later found, in a sham-controlled crossover study using resting-state EEG, increased fronto-parietal coherence in the theta band and a decrease in the gamma band in MCS (n=8) patients following lDLPFC tDCS, while no changes were observed in UWS (n=9) [12]. Another study found decreased central-parietal coherence in the delta band following high-density tDCS over the posterior parietal cortex, along with increased CRS-R total scores in 9 out of 11 DOC patients, even though no significant treatment effect was reported [13]. A sham-controlled randomized double-blind study showed a significant clinical improvement in MCS patients (n=8) following ten sessions of active tDCS targeting the lDLPFC, along with an increase in P300 amplitude [14]. After two weeks of active tDCS over the lDLPFC, Cavinato et al found an increase of power and coherence of the frontal and parietal alpha and beta frequency bands, along with clinical improvements in MCS patients (n=12), whereas UWS (n=12) only showed some local changes in the slow frequencies over frontal electrodes [15]. Finally, Hermann et al recently reported increased power and connectivity in the theta-alpha band after one session of lDLPFC tDCS in responders (12/60 – 20%; 7/32 MCS – 21.9%)  in a prospective open-label study with DOC patients [16]. »

-Regarding the fact that tDCS over the prefrontal cortex modulates brain activity and connectivity even in distant areas, page 9-10 (discussion) read: « The fact that we found an increase of parieto-parietal wSMI and fronto-parietal wPLI connectivity, and considering their ability to capture purely non-linear and a mix of linear and nonlinear interactions respectively, might be explained by a recent theory according to which posterior parts (including parietal regions) of the brain are critical for consciousness, even if fronto-parietal connectivity is still determinant [32]. These authors indeed questioned the role of the frontal cortex in supporting consciousness, suggesting a direct contribution to specific contents of experience from posterior parietal regions, whereas frontal regions rather modulate the neural correlates of consciousness but would not directly contribute to consciousness. In the present study, we found that, even if tDCS was applied over the prefrontal cortex, both fronto-parietal connectivity and connectivity between right and left parietal regions were modulated. »

We have now added a more detailed description about the fact that tDCS over the prefrontal cortex modulates brain activity and connectivity even in distant areas, page 10 now reads: « This is in line with a fMRI study on healthy subjects according to which tDCS would not only modulate the activity under the stimulated area, but also brain network connectivity encompassing long distance brain regions [33]. In DOC patients, the hypothesis is that some long-distance connectivity (encompassing cortical or subcortical regions) may be chronically under-active and that the action of tDCS on cerebral activity and connectivity might counteract this under-activation and induce clinical improvements [10]. »

  1. As the authors already noted, there was no significant difference in brain activity when correcting for multiple comparisons at the group level. This is the main weakness of this study. The authors used a cluster permutation test model at the subject level and found significant effects in the period, session, and interaction. To justify the statistical method, please discuss whether the statistician performed data analysis as an acknowledgement or in the Method section.

Performing statistical analysis at the group level did not lead to significant results when corrected for multiple comparisons. As we have discussed in the paper, this can mainly be due to the small sample size used in this preliminary study. However, to explore other possible reasons for this lack of significant results, we also applied statistical analysis at the subject level to study the single-subject power variation in different sessions and periods. Significant session-period interaction was linked to the application of tDCS. However, generally speaking, the high single-subject variations in the power can be another reason of not seeing significant results at the group level.

Please see below the paragraph discussing this analysis : « Single subject analysis showed that the spectral power variability is significant after both active and sham sessions and even for the interaction of period (i.e. pre-stimulation vs post-stimulation) and session (i.e. sham vs active) for all subjects in all frequency bands. The significant interaction of period and session shows that this variability can be linked to the application of tDCS on the patients. In addition, this high individual variability can be the reason of not seeing significant results in the group level analysis. »

  1. Please check the sentence on lines 291–293 of page 10. Specifically, check to see whether this is an error on line 292: “…following tDCS, after the active tDCS…”

We thank the Reviewer for noticing this error. The sentence has now been corrected and reads: « Our results are therefore in line with the existing literature, showing that increases of alpha and theta power may reflect a higher level of brain activity following active tDCS, even if it is not translated into behavioral responsiveness in all patients after a single session.»

Reviewer 2 Report

This study, albeit concerning a limited number of Patients (9 MCS patients) is well conducted and the analysis of data is of high quality.

I am however a bit skeptical about the clinical usefulness of studies like this.  The reason for my perplexity is that they all lack a “proof of concept” study, in which the EEG effect of prefrontal  tDCS is analyzed in normal subjects.  If we do not know what happens in normals, it is hard to understand what we observe in severely compromised patients.

Nonetheless, I think the study  is worth publishing with only minor revisions required:

  • 4, lines 154-157. Why so few channels were interpolated for EEG analysis (30 to 21) whereas the EEG electrodes were many more (256) ?
  • 9, line 269. The phrase “…three patients displayed a new sign of consciousness only after the active tDCS…” should be modified/mitigated as “…displayed a new sign of reactivity…”.

Author Response

Reviewer #2

This study, albeit concerning a limited number of Patients (9 MCS patients) is well conducted and the analysis of data is of high quality. I am however a bit skeptical about the clinical usefulness of studies like this. The reason for my perplexity is that they all lack a “proof
of concept” study, in which the EEG effect of prefrontal tDCS is analyzed in normal subjects. If we do not know what happens in normals, it is hard to understand what we observe in severely compromised patients. Nonetheless, I think the study is worth publishing with only minor revisions required.

We thank the Reviewer for his positive comments and we totally agree that more proof of concept studies are needed.

1) Pag. 4, lines 154-157. Why so few channels were interpolated for EEG analysis (30 to 21) whereas the EEG electrodes were many more (256)?

Interpolation is an estimation of the power of the electrode based on surrounding electrodes that is only performed during preprocessing when the electrode is too noisy to be included in further steps. Therefore, the (30 to 21) channels the Reviewer is talking about are channels which were too noisy and consequently were rejected at the first level of preprocessing. After performing ICA cleaning, they were interpolated using other electrodes` cleaned signals. In general, as mentioned in line 154, 224 electrodes were used for data analysis purposes.

2) Pag. 9, line 269. The phrase “…three patients displayed a new sign of consciousness only after the active tDCS…” should be modified/mitigated as “…displayed a new sign of reactivity…”.

The sentence has been modified as followed: « Nevertheless, three patients displayed a new behavior associated with consciousness according to the CRS-R only after the active tDCS […] »

Reviewer 3 Report

This is a preliminary study exploring the effects of a single lDLPFC-tDCS session on behavioural and electrophysiological measures of the functioning of a group of DOC patients.  The authors duly report all important findings, also the negative ones, so the reader has an opportunity to draw his own conclusions about results of the study. Though the patient sample is rather small and the reported effects are observed either only on the single subject level or without correction for the multiple testing, the study can be viewed an important step in revealing the complex nature of the tDCS intervention in disorders of consciousness.

There are however a few things that need some clarification. I recommend publication pending minor revisions.

Below is the list of the issues needing more clarification or correction.

page 1, line 45 - "This technique modulates the excitability of targeted brain regions" - a reference documenting this claim would be useful.

page 4, lines 141 & 142 - Please correct "8'th" and "4'th" spelling.

page 5, line 169 - The details of the ANOVA model are unclear, especially the description "epochs power distribution of each session". What was the unit of observation here? A single epoch? A trimmed-mean result from a single channel? Some clarification regarding this issue would be very helpful.

page 6, Table 1 - Please make patient IDs uniform. In this table some of the patient IDs are written without the padding zero.

page 7, Table 2 caption - Please provide explanation what do underlined fonts indicate.

page 8, Figure 3 caption - Please add description what do the white circles indicate, as well as provide the cluster level significance values. Otherwise it is unclear why you present the clusters (indicated by white circles) in this figure. Please remove "-log10(p)" label, since you do not display p-values in a logarithmic format.

page 10, line 293 - The sentence "Single subject analysis showed that the spectral power variability is significant"  is not clear. Please elucidate what does the claim about significance means in this context.

page 10, line 311 - Could you please provide some explanation why alpha-band wPLI connectivity decreased in the sham condition/session? It is not clear why this was observed in this study.

page 13, Appendix A1 - Please correct patient IDs on the figure, making them consistent with the IDs in the main text (the same applies to the Appendices that follow).

page 13, line 387 - Please change "anova" to capital letters (the same applies to Appendices that follow).

page 17, Appendix A7 - As I understand, the asterisk (*) indicates a statistically significant result. In this table this is however not the case. Please correct this. The same comment applies to Appendix A8.

Author Response

Reviewer #3

This is a preliminary study exploring the effects of a single lDLPFC-tDCS session on behavioural and electrophysiological measures of the functioning of a group of DOC patients.  The authors duly report all important findings, also the negative ones, so the reader has an opportunity to draw his own conclusions about results of the study. Though the patient sample is rather small and the reported effects are observed either only on the single subject level or without correction for the multiple testing, the study can be viewed an important step in revealing the complex nature of the tDCS intervention in disorders of consciousness. There are however a few things that need some clarification. I recommend publication pending minor revisions.

We thank the Reviewer for this encouraging comment.

Below is the list of the issues needing more clarification or correction.

-page 1, line 45 - "This technique modulates the excitability of targeted brain regions" - a reference documenting this claim would be useful.

A reference has been added, it now reads: « This technique modulates the excitability of targeted brain regions by inducing a weak electrical current (usually 1-2 mA) between two electrodes (an anode and a cathode) placed on the scalp (Stagg et al, 2018). »

-page 4, lines 141 & 142 - Please correct "8'th" and "4'th" spelling.

The spelling has been corrected.

-page 5, line 169 - The details of the ANOVA model are unclear, especially the description "epochs power distribution of each session". What was the unit of observation here? A single epoch? A trimmed-mean result from a single channel? Some clarification regarding this issue would be very helpful.

To perform ANOVA at the subject level, we considered Session (sham vs active) and Period (pre vs post) as the factors of the model for each subject and fitted the model, at each frequency band separately, to the power values of all single epochs at each electrode. The paragraph has been modified as follows: For the subject level analysis, ANOVA model was fitted to the single-epoch power values of each electrode at different frequency bands considering session (sham vs active) and period (pre vs post) as the factors of the model. In this regard, for each subject and each frequency band the effects of session and period and their interaction were tested. Again, to solve the multiple comparison issue, a cluster permutation test was performed. For all cluster permutation tests, the number of permutations was set to 1000.

-page 6, Table 1 - Please make patient IDs uniform. In this table some of the patients IDs are written without the padding zero.

We thank the Reviewer for noticing this. All the IDs have now been uniformized.

-page 7, Table 2 caption - Please provide explanation what do underlined fonts indicate.

An explanation has been added in the caption of Table 2, it now reads: « The CRS-R total scores in bold and underlined correspond to the total scores obtained by the three responders before and after active tDCS. »

-page 8, Figure 3 caption - Please add description what do the white circles indicate, as well as provide the cluster level significance values. Otherwise it is unclear why you present the clusters (indicated by white circles) in this figure. Please remove "-log10(p)" label, since you do not display p-values in a logarithmic format.

White circles show electrodes of the clusters found in a cluster permutation test. Significant clusters would be colored with red. In this figure, we could not find significant clusters after performing cluster permutation test and that is why the figures are completely in grey. The caption has been modified as follows: Relative power differences in delta, theta, alpha and beta bands in sham and active session as well as for active session compared to sham session. Uncorrected related t-test results are shown for significance level of 0.05. In the permutation cluster test results, white circles show the electrodes of the clusters found in the test. Significant clusters are shown in red (no significant cluster in this figure).”

Lastly, in the 4th column, the values are plotted in logarithmic scales and that is why we wrote -log10(p).

-page 10, line 293 - The sentence "Single subject analysis showed that the spectral power variability is significant" is not clear. Please elucidate what does the claim about significance means in this context.

In this context, we mean statistical significance which was shown using ANOVA and cluster permutation test as explained in the results section (line 217):  « At the subject level, we observed spectral power variability in both sham and active sessions for all subjects in all frequency bands. Cluster permutation test on the F statistics of the ANOVA model found a significant effect of the period, the session and their interaction, on the power values of all subjects at each frequency band. »

The sentence line 293 has been modified for more clarity and now reads: « Single subject analysis showed that there was a high variability in spectral power after both active and sham sessions and even for the interaction of period (i.e. pre-stimulation vs. post-stimulation) and session (i.e. sham vs. active) for all subjects in all frequency bands. The significant interaction of period and session shows that this variability can be linked to the application of tDCS on the patients. In addition, this high individual variability can be the reason of not seeing significant results in the group level analysis. » 

-page 10, line 311 - Could you please provide some explanation why alpha-band wPLI connectivity decreased in the sham condition/session? It is not clear why this was observed in this study.

There is always a level of uncertainty and natural variability for every subject during experiment such as falling sleep, drowsiness, etc. In addition to that, vigilance fluctuations are very common in MCS patients. The main reason of sham session is to capture such variability for each patient to remove its effect from the analysis. In general, we cannot say what exactly caused these reduction of alpha band wPLI connectivity. It can be any of the mentioned reasons. However, this effect was considered in all of our analyses.

Page 9 of the manuscript reads: « Based on these previous results and considering our findings showing that wPLI connectivity in the alpha band was decreased in the sham and increased in the active sessions (which lead to significant increase of fronto-parietal alpha band connectivity), we may argue that anodal tDCS deprived patients` drowsiness in the active session. »

-page 13, Appendix A1 - Please correct patient IDs on the figure, making them consistent with the IDs in the main text (the same applies to the Appendices that follow).

The IDs on the main text and in the tables are now consistent with the IDS on the EEG figures of the Appendices.

-page 13, line 387 - Please change "anova" to capital letters (the same applies to Appendices that follow).

ANOVA have been rewritten in capital letters in all the Appendices.

-page 17, Appendix A7 - As I understand, the asterisk (*) indicates a statistically significant result. In this table this is however not the case. Please correct this. The same comment applies to Appendix A8.

It has been corrected.

Round 2

Reviewer 1 Report

No further comments